# Multidimensional Separation by Magnetic Seeded Filtration: Experimental Studies

**Frank Rhein** *[ID], **Ouwen Zhai, Eric Schmid** and **Hermann Nirschl**

Karlsruhe Institute of Technology (KIT), Institute of Mechanical Process Engineering and Mechanics, Strasse am Forum 8, 76131 Karlsruhe, Germany; ouwen.zhai@kit.edu (O.Z.); eric.schmid@kit.edu (E.S.); hermann.nirschl@kit.edu (H.N.)

* Correspondence: frank.rhein@kit.edu

**Abstract:** The current state of separation technology often neglects the multidimensional nature of real particle systems, which are distributed not only in terms of size, but also in terms of other properties, such as surface charge. Therefore, the aim of this study is to experimentally investigate the applicability of magnetic seeded filtration as a multidimensional separation process. Magnetic seed particles are added to a multisubstance suspension, and a selective heteroagglomeration with the nonmagnetic target particles is induced, allowing for an easy subsequent magnetic separation. The results show that high separation efficiencies can be achieved and that the parameters pH and ionic strength govern the agglomeration process. Selective separation based on surface charge was observed, but undesirable heteroagglomeration processes between the target particles lead to a loss of selectivity. Particle size was clearly identified as a second relevant separation feature, and its partially opposite influence on collision frequency and collision efficiency was discussed. Finally, experimental data of multidimensional separation are presented, in which a size-distributed two-substance suspension is separated into defined size and material fractions in a single process step. This study highlights the need for multidimensional evaluation in general and the potential of magnetic seeded filtration as a promising separation technique.

**Keywords:** multidimensional separation; heteroagglomeration; magnetic seeded filtration; selective separation; surface charge

## 1. Introduction

The field of particle technology has undergone tremendous progress in recent years. So-called engineered nanoparticles [1] are specifically manufactured particle systems that find a broad range of industrial applications due to their desirable properties [2]. Selected examples range from incorporation into electronic devices [3,4], use as a catalyst material [5–7], use in cancer therapy [8–10], to use in wastewater treatment [11–13]. Such applications place high demands on the particle systems: Properties such as particle size, surface, and shape are crucial for correct functionality and, therefore, require precise adjustment, which in turn makes selective separation steps during production a necessity. Furthermore, the release of these particles into the environment poses risks with as yet hard-to-quantify consequences [14–17], which is why effective separation techniques from wastewater streams are necessary. Separation technology has only been able to keep up with this progress to a limited extent. Most separation processes operate on the basis of a single particle feature, as, e.g., filtration separates by particle size. Thus, the separation of particles with an array of defined properties usually requires the series connection of different separation apparatus, leading to inefficient processes and increased energy consumption. Therefore, the identification of multidimensional separation techniques that are able to separate according to several particle features simultaneously is an active field of current research [18–21].

The present work investigates the application of magnetic seeded filtration (MSF) as a selective and multidimensional separation technique that is visualized in Figure 1. Initially, two polydisperse nonmagnetic particle systems with different surface properties are present in the suspension (red and green). Magnetic seed particles (black) are added, and heteroagglomeration is preferentially induced with the red nonmagnetic system due to favorable surface properties. The green particles that do agglomerate tend to be large in diameter, while the fine fraction remains excluded. Formed agglomerates are removed by a following magnetic separation. A subsequent processing step allows, on the one hand, to recover the enclosed nonmagnetic fraction and, on the other hand, to recycle the magnetic seed particles. MSF is thus able to produce particle fractions with defined surface properties (feature 1) and/or sizes (feature 2) from a multicomponent suspension and is therefore considered multidimensional.

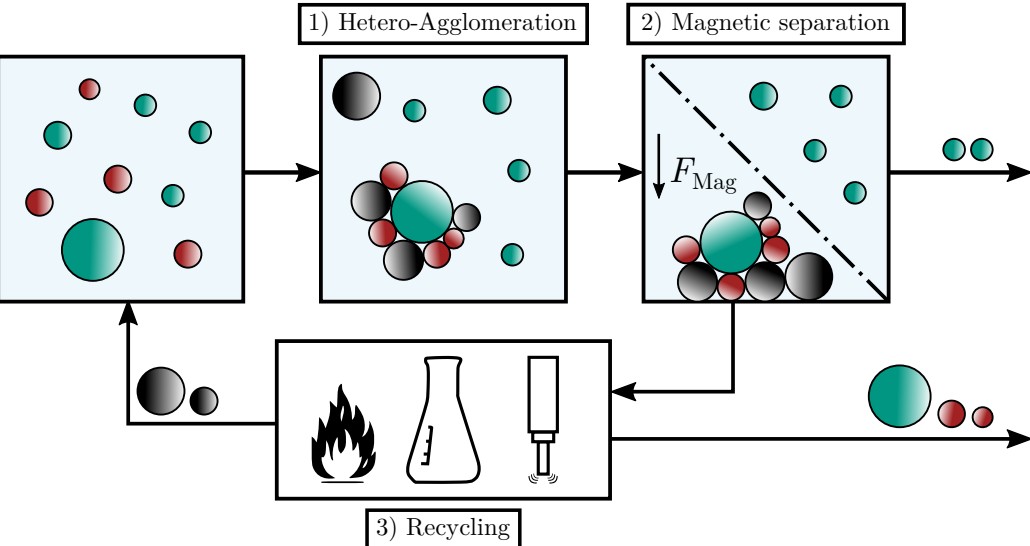

**Figure 1.** Schematic representation of multidimensional magnetic seeded filtration. First, a selective heteroagglomeration is induced (1), and formed agglomerates are separated due to their newly acquired magnetic properties (2). Separated agglomerates may be broken up to recycle the magnetic component and gain access to the separated nonmagnetic particles (3).

Different designations for ultimately the same separation principle are found in the literature, as, e.g., *magnetic flocculation* [22], *magnetic adsorption* [23], and as used here, *magnetic seeded filtration* [24]. This inconsistency in nomenclature may be partially responsible for MSF's relative unfamiliarity in the separation community despite various promising application studies, which are, e.g., summarized by Franzreb [23]: From use in the harvesting of microalgae [25], the separation of phosphates from wastewater [26], the removal of oil leakages [27], to the purification of proteins [28,29], MSF shows a broad application spectrum. MSF was further used for the purification of wastewater from chemical-mechanical polishing [30], highly turbid groundwater [31], or the separation of nonmagnetic particles during the purification of gear oils [32], while latest research even shows that MSF is applicable to the separation of microplastics from environmental samples [33–35] and is able to achieve high selectivity. Although the number of relevant publications has been increasing exponentially over the last decades, no industrial-scale applications have been realized so far besides the MIEX technology for water purification [23,36]. This is astounding since MSF offers some distinct advantages compared with more popular separation techniques: As the magnetic force responsible for separation is exceptionally large for the right choice of magnetic seed material, separation matrices are designed openly, which leads to lower pressure loss compared with, e.g., conventional cake or depth filtration [37]. Furthermore, MSF is generally not limited to a certain particle size range, as agglomeration processes occur on nearly every scale and are especially dominant for smaller particle sizes. Here,

traditional separation techniques, such as cake filtration or centrifugation, tend to struggle. Additionally, high separation efficiencies are achieved even at low volume fractions of the magnetic component [35], and the required magnetic field can be set up economically through the use of permanent magnets [38,39]. These benefits make magnetic separation applicable in a wide range of applications [40,41].

One of the main points of criticism against MSF is the necessary addition of magnetic seed material, which must be recycled and reused for the process to be both economically and ecologically viable. The necessary agglomerate breakup is, e.g., studied via dissolution of the separated component [42,43] via desorption [44] or via mechanical breakup [45]. In a previously published study [46], thermal, chemical, and mechanical breakup were investigated with respect to the recycling rates of both magnetic and nonmagnetic material as well as the functionality of the magnetic seed particles over the course of multiple process cycles. To the authors' best knowledge, this is the most holistic perspective on agglomerate breakup and recycling during MSF. It showed high recycling rates and general feasibility for all three approaches, while also discussing the respective benefits, limitations, and possible applications.

With regard to application as a selective and multidimensional separation process, two questions remain unanswered after intensive literature research: The size dependence of the separation has not yet been investigated or discussed satisfactorily. Furthermore, almost all publications are concerned with the separation of a single nonmagnetic particle system, where the objective is usually the highest possible separation efficiency. The case of a multicomponent system, where selectivity is most essential, has been disregarded as of yet. This study aims at closing these gaps.

## 2. Materials and Methods

### 2.1. Magnetic Seeded Filtration Theory

The efficiency of MSF is governed by the heteroagglomeration between magnetic and nonmagnetic target particles. The kinetics of a general heteroagglomeration process between the two particles $i$ and $j$ can be expressed as

$$\frac{\mathrm{d}N_{ij}}{\mathrm{d}t} \propto N_i N_j \beta_{i,j} \alpha_{i,j}, \tag{1}$$

where the number concentrations $N_i$ and $N_j$ and two kinetic parameters, $\beta$ and $\alpha$, are decisive.

The collision frequency $\beta$ quantifies the amount of collisions between the particles. In the orthokinetic case, i.e., when collision is governed by fluid flow rather than diffusion, $\beta$ can be approximated according to

$$\beta_{i,j} = \frac{4}{3}\left(r_i + r_j\right)^3 \bar{G} \tag{2}$$

Ref. [47]. It is apparent that $\beta$ strongly depends on the particle radii $r$ and the mean shear rate in the system $\bar{G}$.

A collision in itself is not sufficient for two particles to agglomerate, as they also have to "stick", the probability of which is expressed by the collision efficiency $\alpha$, which shows the following dependencies:

$$\alpha_{i,j} \propto \underbrace{\exp\left(-C_1\left(1 - \frac{r_i}{r_j}\right)\right)\left(r_i r_j\right)^{-C_2}}_{(a)} \underbrace{\exp\left(-\frac{E_{\Sigma,max}}{k_B \vartheta_A}\right)}_{(b)} \tag{3}$$

In the diffusion-controlled (perikinetic) case, $\alpha$ is derived from geometrical parameters but mainly from an integration of the resulting interaction potential $E_{\Sigma,i,j}(h)$ between the

particles [48]. This work is only concerned with the classical DLVO theory, namely, that electrostatic and van der Waals interactions are superimposed according to

$$E_{\Sigma,i,j}(h) = E_{\text{el},i,j}(h) + E_{\text{vdW},i,j}(h). \tag{4}$$

Note that for other particle systems, the consideration of hydrophobic interaction energies may be required [34,35]. The electrostatic interaction energy $E_{\text{el},i,j}$ is defined as [49]

$$E_{\text{el},i,j}(h) = \frac{128\pi r_i r_j N_A k_B T}{(r_i + r_j)\kappa^2} \gamma_i \gamma_j \exp(-\kappa h) \tag{5}$$

$$\kappa^{-1} = \sqrt{\frac{\varepsilon k_B T}{2e^2 I N_A}} \quad \Big| \quad \gamma_k = \tanh\left(\frac{e\zeta_k}{4k_B T}\right) \tag{6}$$

and represents the principal way of influencing the the interaction energy. The reciprocal DEBYE length $\kappa$ is a measure for the range of the diffusive double layer and, therefore, strongly influences the range of electrostatic interaction. The absolute strength is given by the dimensionless surface potentials $\gamma$ of the particles and is quantified by the measurable zeta potential $\zeta$. Electrostatic interactions may be attractive or repulsive depending on whether or not the zeta potentials show opposite signs. The van der Waals interaction is always attractive and expressed in simplified form according to [47]

$$E_{\text{vdW},i,j}(h) = -\frac{A_{H,i,j} r_i r_j}{6h(r_i + r_j)}, \tag{7}$$

with the main influencing parameters being the HAMAKER constant $A_{H,i,j}$ and the particle radii. When estimating the collision efficiency, the largest contribution stems from a region close to the maximum in the interaction energy curve $E_{\Sigma,i,j}(h)$. Therefore, the perikinetic collision efficiency may be estimated by the height of the energy barrier $E_{\Sigma,i,j,\text{max}}$, which is shown by factor (b) in Equation (3) [47,50].

For orthokinetic agglomeration, the calculation of $\alpha$ is more complicated, as flow effects have to be taken into account, which usually lead to a reduction of the agglomeration probability. Empirical equations derived by trajectory analyses [51] or simulation [52] offer a simple way of estimating the influence of these flow effects and, consequently, of the particle size on the collision efficiency, which is incorporated by Factor (a) in Equation (3) [51]. The collision efficiency is reduced for large differences in particle size ($r_i/r_j \ll 1$) and overall large particles ($(r_i r_j)^{-C_2}$). $C_1$ and $C_2$ are empirical parameters determined by experiments. This approach allows for a clear separation between flow effects and particle-particle interaction, while simultaneously integrating the influence of particle size on collision efficiency.

After heteroagglomerates are formed, they are separated due to their newly gained magnetic properties. A larger magnetic force

$$F_{\text{Mag}} = \mu_0 V_M M \nabla H \tag{8}$$

is achieved by an increased partial volume of the magnetic component $V_M$, an increased magnetization $M$ of the magnetic material, or increased magnetic field gradients $\nabla H$.

### 2.2. Particle Systems

The inorganic particle systems $SiO_2$ (SF800/SF600) and ZnO (Silatherm) were are used as a nonmagnetic component. Both systems were purchased from Quarzwerke GmbH and are produced from prepared natural raw minerals. A $SiO_2$–magnetite composite material, $SiO_2$–MAG (AR1062), purchased from Microparticles GmbH is used as a magnetic particle system. Both $SiO_2$ products (SF800 and SF600) are of identical material of the same manufacturer, with the sole difference being the slightly different grind size.

Figure 2 shows the zeta potentials determined by *Zetasizer Nano ZS* (Malvern Panalytical) at a constant ionic strength $I = 0.1\,\text{M}$ and particle concentration $c_V = 3.75 \cdot 10^{-3}\,\text{vol}\%$ for different $pH$ values. As the sedimentation of SF600 leads to inaccurate measurements, the values shown for $SiO_2$ were measured with SF800 and are assumed to also be valid for SF600. The zeta potentials of all presented material systems decrease with an increasing $pH$ value, but show characteristic differences: $SiO_2$ has no isoelectric point (IEP, $\zeta = 0$) over the entire $pH$ range, while $SiO_2$–MAG has an IEP at about $pH = 4$ and a positively charged surface at lower $pH$ values. At high $pH$ values, $SiO_2$ and $SiO_2$–MAG show almost identical behavior at $\zeta = -30\,\text{mV}$. ZnO is generally less charged at high $pH$ and has an IEP at about $pH = 8$. For acidic conditions, no data are shown for ZnO particles, as they are dissolved.

The PSDs in Figure 3 clearly show that the magnetic particles are monodisperse, while the nonmagnetic particle systems are polydisperse. Measurements for $SiO_2$–MAG were performed by analytical ultracentrifugation (AUC) with the disc centrifuge *DC24000* (CPS Instruments), while all other particle systems were analyzed by laser diffraction (*Helos*, Sympatec). Figure 3 underlines that $SiO_2$ (SF800) and ZnO show almost identical PSDs, while $SiO_2$ (SF600) shows slightly higher particle diameters.

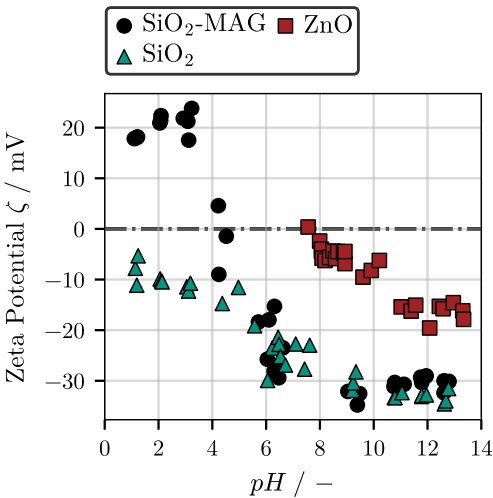

**Figure 2.** Zeta potentials of the used particle systems measured with *Zetasizer Nano ZS* (Malvern Panalytical) at a constant ionic strength $I = 0.1\,\text{M}$ and particle concentration $c_V = 3.75 \cdot 10^{-3}\,\text{vol}\%$.

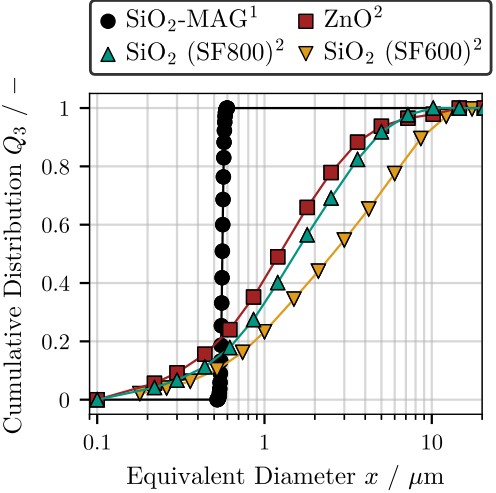

**Figure 3.** PSDs of the used particle systems. [1] Measured by AUC (*DC24000*, CPS Instruments). [2] Measured by laser diffraction (*Helos*, Sympatec).

### 2.3. Experimental Procedure and Parameters

MSF experiments were performed similarly to [53]. Initially, stock suspensions of the dry particle systems were prepared by sonification with *Digital Sonifier 450* (Branson Ultrasonics). A sample of the stock suspension $P_0$ was taken for later analysis, in order to take deviations in preparation into account. Subsequently, the respective volumes of the stock suspensions to achieve the desired volume concentrations $c_{v,i}$ were transferred into a snap-on lid vial. The suspension parameters ionic strength $I$ and $pH$ were adjusted by the addition of 2 M NaCl solution and 0.5 M HCl or NaOH solution, respectively. Finally, the experimental volume was filled to $V_L = 30$ mL with ultrapure water. The suspension was agitated for the agglomeration time $t_A$ in the laboratory shaker *Vortex Genius 3* (IKA GmbH). After agglomeration, a ferromagnetic separation matrix was immersed in the suspension, and the vial was positioned inside a dipolar openable Halbach magnet (permanent magnet) with a magnetic flux density of $B = 0.2$ T [39]. Magnetic separation was performed for $t_S = 2$ min, which was shown in preliminary studies to guarantee full separation of pure magnetic suspensions, while purely nonmagnetic suspensions did not show any measurable separation efficiency. This shows that the separation of nonmagnetic particles can be entirely attributed to the heteroagglomeration. A representative sample $P_E$ was taken from the nonseparated suspension and sonified again in order achieve comparable levels of dispersity. The samples $P_0$ and $P_E$ were analyzed via UV–VIS spectroscopy and AUC (see Sections 2.5–2.7). All experimental parameters are summarized and linked to their corresponding figure of Section 3 in Table 1.

**Table 1.** Relevant parameters of the experimental studies with corresponding figures of Section 3 showing the results.

| | NM1 | NM2 | $c_{v,0,M}$ [vol%] | $c_{v,0,NM1}$ [vol%] | $c_{v,0,NM2}$ [vol%] | $t_A$ [min] |
|---|---|---|---|---|---|---|
| Figure 5 | SiO$_2$ (SF300) | – | $2.8 \cdot 10^{-3}$ | – | $3.75 \cdot 10^{-3}$ | 10 |
| Figure 5 | ZnO | – | $2.8 \cdot 10^{-3}$ | $3.75 \cdot 10^{-3}$ | – | 10 |
| Figure 6a,b | SiO$_2$ (SF300) | ZnO | $2.8 \cdot 10^{-3}$ | $3.75 \cdot 10^{-3}$ | $3.75 \cdot 10^{-3}$ | 10 |
| Figure 6c | SiO$_2$ (SF300) | ZnO | $2.8 \cdot 10^{-3}$ | $3.75 \cdot 10^{-3}$ | var. | 10 |
| Figure 8 | SiO$_2$ (SF600) | – | $5.6 \cdot 10^{-3}$ | $1.5 \cdot 10^{-2}$ | – | 5 |
| Figure 9 | SiO$_2$ (SF600) | ZnO | $5.6 \cdot 10^{-3}$ | $1.5 \cdot 10^{-2}$ | $7.14 \cdot 10^{-3}$ | 5 |

### 2.4. Evaluation of a Separation Experiment

The overall separation success of a nonmagnetic component $i$ is quantified via the separation efficiency

$$A_{NM,i} = 1 - \frac{m_{i,E}}{m_{i,0}} = \frac{m_{i,SEP}}{m_{i,0}}. \tag{9}$$

The absolute masses $m_i$ are calculated from the volume concentrations $c_{v,i}$, sample volume $V$, and material density $\rho_i$. When information about the PSD before and after the experiment is known, the grade efficiency

$$T_{NM,i}(x) = 1 - (1 - A_{NM,i}) \frac{q_{3,i,E}(x)}{q_{3,i,0}(x)} \tag{10}$$

quantifies the size dependence of the process. $A$ and $T$ are defined analogously, as a value of 1 represents full and a value of 0 indicates no separation at all.

If a suspension contains more than one nonmagnetic component, the selectivity of the separation is of special importance. It provides information on how the separation efficiencies of the respective materials behave relative to each other. In mechanical process engineering, the term selectivity is only associated with sharpness of separation according to particle size during classification. However, there is no consistent definition for a material-specific selectivity that is required for describing separation experiments in multicomponent systems. In chemical engineering, the selectivity of a reaction with respect to the product $i$

is defined by the ratio of the substance *i* formed and the total amount of reacted substance. Transferring this definition to separation processes, the mass-based selectivity $S_{m,i}$ with respect to the component *i* is interpreted as the ratio of the separated mass of the component *i* to the total separated mass. Equation (11) derives an expression for the said selectivity with respect to the separation efficiencies *A* in a suspension with *N* components.

$$S_{m,i} = \frac{m_{i,\mathrm{SEP}}}{\sum\limits_{n=1}^{N} m_{n,\mathrm{SEP}}} = \frac{m_{i,\mathrm{SEP}}}{\sum\limits_{n=1}^{N} A_{\mathrm{NM},n} m_{n,0}} \cdot \frac{m_{i,0}}{m_{i,0}} = \frac{A_{\mathrm{NM},i} X_{m,i,0}}{\sum\limits_{n=1}^{N} A_{\mathrm{NM},n} X_{m,n,0}} \tag{11}$$

$$X_{m,i,0} = \frac{m_{i,0}}{\sum\limits_{n=1}^{N} m_{n,0}} \tag{12}$$

*2.5. UV–VIS Analysis*

When light of known intensity $I_0$ is transmitted through a medium, its intensity decreases, i.e., for the intensity after the sample, $I < I_0$ applies. The Beer–Lambert law

$$E_\lambda = \log_{10}\left(\frac{I}{I_0}\right)_\lambda = d\sum_{n}^{N} c_n k_{n,\lambda} \tag{13}$$

relates the extinction $E_\lambda$ at each wavelength $\lambda$ that includes both absorption and scattering of light to the length of the irradiated path *d*, the concentrations of dissolved or suspended substances $c_n$, and their wavelength-specific extinction coefficients $k_{n,\lambda}$ [54]. Extinction coefficients were determined via the linear regression of dilution series data for each material *n*. If all $k_{n,\lambda}$ are known, the concentrations of a multicomponent suspension are determined from a measured extinction spectrum $\hat{E}_\lambda$ via least squares optimization of Equation (13) [55]. All measurements were performed with the UV–VIS spectrometer *FLAME-S-XR1-ES* (Ocean Insight).

*2.6. AUC Analysis*

AUC analysis was performed in the disc centrifuge *DC24000* (CPS Instruments). To ensure stable sedimentation, a radial density gradient $\rho_\mathrm{L}(r)$ is established within the rotating volume. For this purpose, identical volumes of sugar solution with a decreasing mass concentration from 24 to 8 w% are injected during rotation at a constant rotation frequency $\omega$. When injecting a sample, the particles sediment from the initial radial position $r_0$ outwards, until they reach a single-wavelength extinction detector at $r_\mathrm{D}$. Assuming perfect spheres, the equivalent diameter

$$x = \sqrt{\frac{18\eta}{(\rho_\mathrm{F} - \rho_\mathrm{L})\omega^2 t_{\mathrm{sed}}} \ln\left(\frac{r_\mathrm{D}}{r_0}\right)} = \sqrt{\frac{C_{\mathrm{sed}}}{t_{\mathrm{sed}}}} \tag{14}$$

is related to the required sedimentation time $t_{\mathrm{sed}}$ required to reach the detector. All material and process conditions can be summarized into the constant $C_{\mathrm{sed}}$, which is determined via the measurement of a known calibration standard. The amount of particles in each size class is determined via extinction analysis, similar to Equation (13).

*2.7. Multidimensional Analysis*

The evaluation of a multidimensional separation experiment requires the determination of both the concentrations and PSDs in multicomponent suspensions. As shown in Equation (13), extinctions are generally additive. By measuring the extinction spectrum and evaluating it at more than one wavelength, the concentrations sought can be determined as described in Section 2.5. However, it is necessary that the extinction coefficients $k_{n,\lambda}$ differ sufficiently. Substance-independent scattering effects often dominate the extinction spectrum of suspensions, which usually show only a weak wavelength dependence. This results in similar spectra between the materials, which makes a precise evaluation

of individual concentrations difficult. The determination of the PSD is limited by the fact that the disc centrifuge only measures the extinction at one wavelength ($\lambda = 470$ nm). In a two-component suspension, this results in an underdetermined system of equations (see Equation (13)), and the measured extinction cannot be clearly attributed between the respective components. An analysis is only possible if both material systems differ in their particle size and/or density in such a way that they arrive at the detector separately in time. In general, however, this cannot be assumed.

To mitigate these limitations, a component reduction approach was developed, validated, and applied within the scope of this work. First, a sample is divided into four equal subsamples. One is used for UV–VIS and one for AUC analysis directly. Subsequently, the $pH$ value of the other samples is lowered to $pH < 2$ by the addition of HCl solution, whereby the resulting dilution is taken into account during later evaluation. During zeta potential measurements, ZnO has been shown to be dissolved in acidic conditions. Such a solution shows no extinction in the investigated wavelength range, which ultimately means that ZnO is masked for spectral analysis by lowering the $pH$ value. In preliminary investigations with pure $SiO_2$ suspensions, it was also confirmed that the low $pH$ value has no influence on the extinction properties of $SiO_2$. It follows that an analysis of the acidic multicomponent suspension is equivalent to a single-component suspension containing only $SiO_2$ in identical concentration and PSD, which can therefore be determined by the described methods. If only two components ($SiO_2$ and ZnO) were originally present, the signal of the pure ZnO particles is obtained by subtracting the acidic extinction signal from the neutral one. From these data, the concentration and PSD of the contained ZnO particles can then be determined. Figure 4 shows this procedure using a concrete AUC example and illustrates the additivity of the extinctions, since the individual peaks can be guessed at in the neutral sample, but are reliably split by the component reduction method.

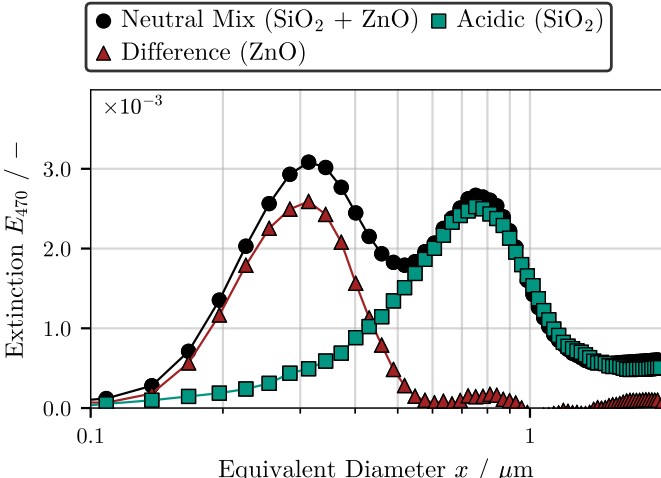

**Figure 4.** Single-wavelength extinction over equivalent diameter during AUC analysis of a multi-component suspension containing both $SiO_2$ and ZnO. Measured signals are shown without (neutral) and with (acidic) addition of HCl. Additionally, the difference of both signals is plotted.

## 3. Results and Discussion

### 3.1. Separation Based on Surface Charge

Data shown in this chapter were already published and discussed in [56,57]. However, as it is the foundation for the following selective and multidimensional separation experiments, discussion is repeated here. Figure 5 shows separation efficiencies for either $SiO_2$ (SF800) or ZnO at different values for $pH$ and $I$. For both $pH$ values shown, the separation efficiency of $SiO_2$ increases with increasing ionic strength. Since both $SiO_2$ and $SiO_2$–MAG are negatively charged in the investigated $pH$ range (see Figure 2), the electrostatic interactions are repulsive and counteract agglomeration. According to Equation (6), an increase

in ionic strength leads to a reduction in DEBYE length, a reduced range of electrostatic repulsion, and thus an increase in collision and separation efficiency. Similarly, for lower absolute zeta potentials, i.e., at $pH = 7$ versus $pH = 12$, the reduced strength of repulsive interaction also leads to an increase in separation efficiency. Regarding ZnO, an increased separation compared with $SiO_2$ is evident for the parameter range presented. Based on the same discussion, this is to be expected in view of Figure 2, because ZnO exhibits a lower absolute zeta potential value compared with $SiO_2$. Consequently, the repulsive electrostatic interaction between ZnO and the magnetic particles is lower and the separation efficiency consequently higher. These differences in zeta potential also lead to a different ionic strength dependence: For $pH = 7$, the ZnO system is already completely destabilized and thus independent of the ionic strength. For $pH = 12$, no separation of ZnO occurs at $I = 0.01$ M, but an increase to $I = 0.1$ M is sufficient to destabilize the system again. With regard to the desired selective separation, Figure 5 highlights the promising parameter combination $pH = 12$, $I = 0.1$ M. Here, the separation efficiencies of $SiO_2$ and ZnO differ significantly, and a selective separation of ZnO is expected. Figure 5 underlines the general potential of MSF: Only little effort and material input are required to achieve an almost complete separation for ZnO. Especially the suitability for diluted suspensions and small particle size ranges distinguishes MSF from known methods, such as filtration or centrifugation.

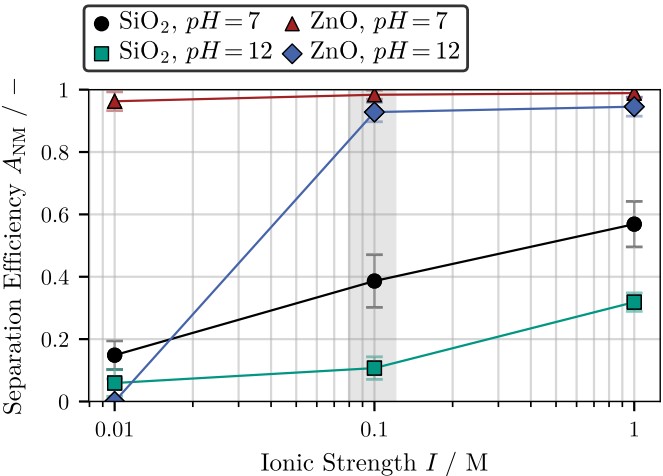

**Figure 5.** Separation efficiencies of either $SiO_2$ (SF800) or ZnO at different values of $pH$ and ionic strength $I$.

### 3.2. Selectivity Based on Surface Charge

Parts (a) and (b) of Figure 6 show experimental results for the selectivity of the process with respect to ZnO ($S_{ZnO}$) for different combinations of $pH$ and $I$. The selectivity defined in Equation (11) equals 1 for the case that no $SiO_2$ and 0.5 for the case that $SiO_2$ and ZnO are separated equally. A selectivity below 0.5 means an increased separation of $SiO_2$ and was not observed in this study. In addition to the experimental values, predictions of selectivity based on the results in Section 3.1 are shown. These are based on the assumption that the separation efficiencies shown in Figure 5 are transferable to the multicomponent system, i.e., that $SiO_2$ and ZnO are separated identically to Figure 5. The highlighted parameter setting $I = 0.1$ M, $pH = 12$ is included in both parts of Figure 6 and shows an experimental selectivity of $S_{ZnO} > 0.8$, proving that a selective separation between ZnO and $SiO_2$ occurs, as the separation efficiency of ZnO is higher than that of $SiO_2$ by more than a factor of 4 (see Equation (11)). Based on the results from Section 3.1, this is understandable, as ZnO has a lower zeta potential than $SiO_2$, therefore a lower electrostatic repulsion towards the magnetic particles and an enhanced separation efficiency. A reduction in $pH$ to 7 leads to a drastic decrease in selectivity, as shown in Figure 6a. The same applies to an increase in

ionic strength to $I = 1\,\text{M}$ shown in Figure 6b. In both cases, only a selectivity of $S_{ZnO} < 0.6$ is obtained, which means that ZnO and $SiO_2$ are separated similarly. Both effects are to be expected based on Figure 5, as both a reduction in *pH* and an increase in ionic strength do not significantly affect the separation efficiency of ZnO particles, as they are almost fully separated for all investigated parameter settings. However, both changes lead to an increase in the separation efficiency of $SiO_2$ due to the reduced range and strength of the electrostatic interaction and the associated increase in agglomeration discussed in Section 3.1. If the separation efficiency of $SiO_2$ increases while that of ZnO remains constant, the observed negative trend in selectivity follows.

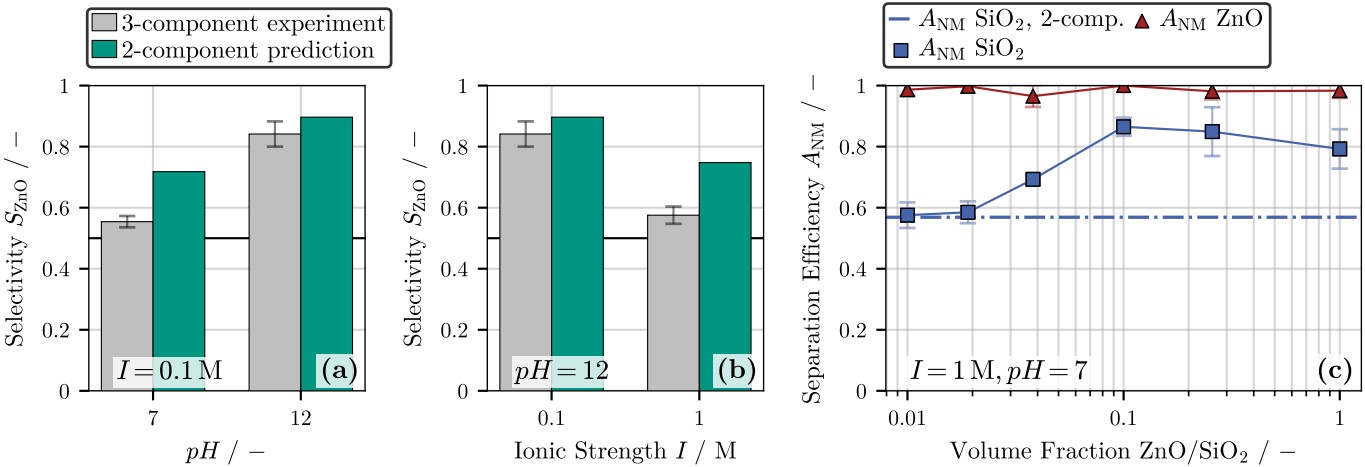

**Figure 6.** (**a**,**b**): Selectivity towards ZnO for varying *pH* and ionic strength *I*. Both the predicted values based on Figure 5 and experimental values are shown. (**c**): Experimental separation efficiencies of $SiO_2$ and ZnO for a varying volume fraction of nonmagnetic particles in the multicomponent suspension.

However, Figure 6 further shows that the experimental value is consistently below the predicted selectivity. Specifically, this means that $SiO_2$ is separated at a higher rate in the multicomponent system. As identical volume concentrations of $SiO_2$ and ZnO are used with a constant amount of magnetic material, the overall volume ratio between magnetic and nonmagnetic particles is reduced in the multicomponent system (see Table 1). If relevant, this should lead to a reduced separation efficiency of $SiO_2$ and oppose the observation. The selectivity difference between experiment and prediction is small for the parameter setting $I = 0.1\,\text{M}$, $pH = 12$, but increases as soon as either the *pH* value is lowered or the ionic strength is increased. All other experimental parameters being equal, this discrepancy must be due to an interaction between the ZnO and $SiO_2$ particles. Such an effect was already postulated and theoretically investigated in a previous study [53] and is now experimentally proven for the first time. Figure 7 visualizes the initial stages of the agglomeration process and provides the explanation for the observed phenomenon. Initially, i.e., when only primary particles are present, $SiO_2$ agglomerates poorly, while ZnO agglomerates strongly with magnetic particles ($k_{1,1} = 0$ and $k_{1,2} > 0$), which leads to the observed differences in separation behavior shown in Figure 5. In the multicomponent system, an agglomeration between $SiO_2$ and ZnO is possible and also likely, when comparing the zeta potentials in Figure 2 ($k_{1,3} > 0$). The heteroagglomerates formed in this way do not contain any magnetic material, are therefore not separated, and are ultimately not directly responsible for the observed drop in selectivity. However, their surface is heterogeneous and contains both regions of $SiO_2$ and ZnO. Therefore, it is to be expected that in later agglomeration steps, three-component agglomerates are also formed. These heteroagglomerates are separated due to the contained magnetic material, and $SiO_2$ particles are ultimately separated, although they do not engage in direct agglomeration with the magnetic component. Vividly, ZnO acts as a kind of flocculant for $SiO_2$ and the

magnetic particles. This explains the increased separation of SiO$_2$ and the lower selectivity compared with the predictions on the basis of ZnO-free experiments.

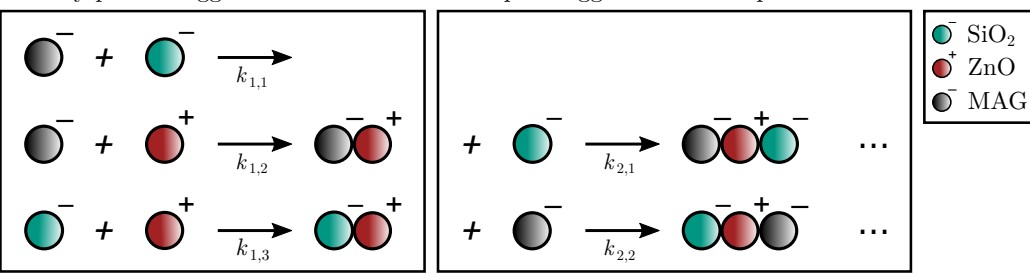

**Figure 7.** Schematic representation for selectivity loss in charge-based separation of multicomponent suspensions.

In order to test this hypothesis experimentally, another experimental series was carried out, varying the volume concentration of the ZnO particles $c_{v,ZnO}$ while keeping the concentration of SiO$_2$ constant, resulting in variations of the volume ratio between ZnO and SiO$_2$ (see Table 1). The results are shown in Figure 6c. As is expected, complete separation of ZnO occurs. Starting from a volume ratio of 1, the separation efficiency of the SiO$_2$ particles is initially constant for decreasing ZnO concentrations, but decreases significantly for volume ratios $< 0.1$. For very low amounts of ZnO, the separation efficiency of SiO$_2$ approaches the measured value from Figure 5, which is visualized by the horizontal line. This clearly proves that the measured selectivity loss in the multicomponent system is due to hetero-agglomeration between the nonmagnetic particles. Although this effect is undesirable for selective separation, it can be useful in other applications: Figure 7 shows that the separation efficiency of a material system, which by nature does not, or only slightly, undergo heteroagglomeration processes with the magnetic particles, can already be significantly increased by the addition of small amounts of opposing or uncharged particles. If, for example, in a clarification application, the goal is to produce a SiO$_2$-free liquid phase, the process result is improved by the addition of ZnO particles. Under certain circumstances, such an addition can make the adjustment of the suspension parameters $pH$ and $I$ obsolete and potentially lower process costs.

### 3.3. Selectivity Based on Particle Size

Figure 8 shows the results for the grade efficiency in an experiment with SiO$_2$ (SF600). Only every 50th data point is shown, and the standard deviation (triple determination) is indicated by the gray area. As grade efficiency is clearly increasing, Figure 8 shows that larger SiO$_2$ particles are separated more effectively than smaller ones in the investigated parameter range. At first glance, the error range is large, but almost constant for all particle diameters. This is due to the fact that the individual measurements differ in their absolute values of $T_{NM}$, but each shows the same general trend of increasing grade efficiency. It should be noted that as $T_{NM}$ increases, the particle concentration and extinction of the respective diameter class in the sample after separation $P_E$ decreases. Additionally, larger particles generally show lower extinction coefficients, resulting in further reduction of the extinction signal for larger particles. To minimize the resulting uncertainties, only data with a signal-to-noise ratio of at least 2 are used for the determination of the grade efficiency. Most data points, especially in the medium particle size range, exceed this value many times over. In addition, preliminary tests were carried out without the addition of magnetic particles under otherwise identical conditions. As they showed no change in PSD, an apparent grade efficiency due to shifting particle sizes caused by homoagglomeration is ruled out. It is therefore concluded that the shown increase in $T_{NM}$ with particle size is significant.

The discussion of particle size dependence is by no means trivial, as all factors of magnetic separation and agglomeration kinetics discussed in Section 2.1 are affected.

Concerning the magnetic separation, Equation (8) shows that an increasing magnetic particle size results in a higher absolute magnetic force per particle. However, as inertia during separation is also increased, an earlier study [56] derived a separation criterion that showed that the magnetic separation of a heteroagglomerate is mainly influenced by the volume ratio between the magnetic and nonmagnetic component. It should be noted that in this study, the size of the magnetic seed particles was not varied; however, it is sensible to assume that larger nonmagnetic particles are more likely to form agglomerate with a lower magnetic partial volume and might therefore be separated to a lesser extent. Nevertheless, the influence of particle size on the separation efficiency is ultimately determined by the agglomeration kinetics: On the one hand, an increased particle diameter leads to an increased collision frequency in the orthokinetic range, which is due to the larger collision radii (see Equation (2)). At the same time, an increased particle size at a constant volume concentration results in a decreasing particle number concentration, which in turn has a negative effect on the total number of agglomeration events per unit time (see Equation (1)). Both the van der Waals and electrostatic interaction energies increase proportionally to the particle diameter $x$. At the same time, however, inertial forces and gravity, which counteract agglomeration, increase proportional to $x^3$ and outweigh particle-particle interactions for large particles. These effects generally lead to a reduction of collision efficiency with increasing particle size and are accounted for by term (a) of Equation (3). Additionally, the particle size ratio $r_i/r_j$ plays an important role: For particles of unequal size, the probability of agglomeration is decreased due to flow phenomena. Since the magnetic particles are monodisperse and significantly smaller than the $SiO_2$ system, this particle size ratio decreases with the increasing size of the nonmagnetic particles and leads to a reduced collision efficiency according to Equation (3). Ultimately, the interplay of all these partially opposing influences determines whether the agglomeration rate, and thus the grade efficiency, increases or decreases with increasing particle size. However, it is unlikely that all effects will exactly cancel each other out and that the MSF process will be independent of particle size. Since the balance between collision efficiency and collision frequency strongly depends on the respective system or the set parameters, a general statement on how the grade efficiency of MSF behaves with the increasing particle size is not permissible. This is underlined by the fact that in an earlier study [53], grade efficiency decreased with particle size, whereas in the parameter field considered here, the positive effect on the collision frequency outweighs the negative effect on the collision efficiency and the particle number concentrations, and an increase in grade efficiency with increased particle size is observed.

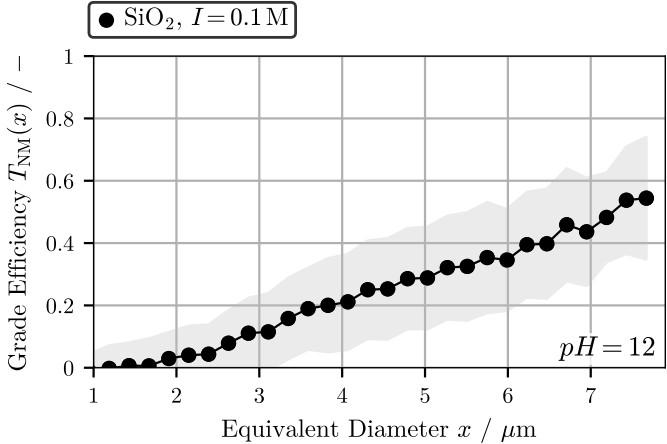

**Figure 8.** Grade efficiency curve for $SiO_2$ (SF600) at a constant ionic strength and $pH$. The gray shaded area represents the standard deviation of triple determination.

### 3.4. Multidimensional Separation

Figure 9 shows the results of two multidimensional separation experiments, where both the concentrations and PSDs before and after separation were measured. As already discussed above, the AUC analysis reaches a detection limit in the case of large particles and/or high separation efficiencies. As ZnO particles are almost completely separated regardless of their particle size, no significant extinction signal was measurable after separation during AUC. Therefore, separation efficiencies of the ZnO particles were determined by means of UV–VIS analysis and are shown as a horizontal line in Figure 9. Furthermore, the grade efficiency of the SiO₂ particles is only shown for data points with a signal-to-noise ratio greater than 2. This is evident from the fact that both grade efficiency curves stop at a certain particle size, even though the measurement is performed on the entire particle size range. For larger particles, no significant extinction signal is measurable after separation and complete separation can be assumed. In agreement with Figure 8, the grade efficiency of the SiO₂ particles increases with increasing particle size, although the absolute values of the grade efficiency are significantly higher in the multicomponent system. This is due to the heteroagglomeration of SiO₂ with ZnO, which was discussed in Section 3.2. It can be seen that a reduction in ionic strength leads to a reduction in grade efficiency, which is expected due to the increased DEBYE length. It is interesting to note, however, that the dependence on particle size is not significantly influenced by this. Although the grade efficiency curves are offset, their slopes are almost identical. This suggests that the influences of particle size and surface are independent from each other. This is a crucial detail with regard to the application of multidimensional MSF, as it implies that both dependencies can be set individually and allow for a flexible way of defining the separated fractions. In the example shown here, the grade efficiency curve can be shifted towards larger particle diameters by reducing the ionic strength without altering the relative classification with respect to particle size.

Vividly, Figure 9 shows that it is possible to classify a multicomponent suspension of polydisperse ZnO and SiO₂ particles according to the particle features size and charge in a single process step by means of MSF. In the separated fraction, all ZnO and predominantly large SiO₂ particles are found, while only the fine fraction of the SiO₂ system is found in the nonseparated fraction. Even though both the selectivity with respect to particle size and surface charge are far from ideal, the presented results underline the versatility and potential of MSF as a new and as yet unestablished separation technique in multidimensional separation.

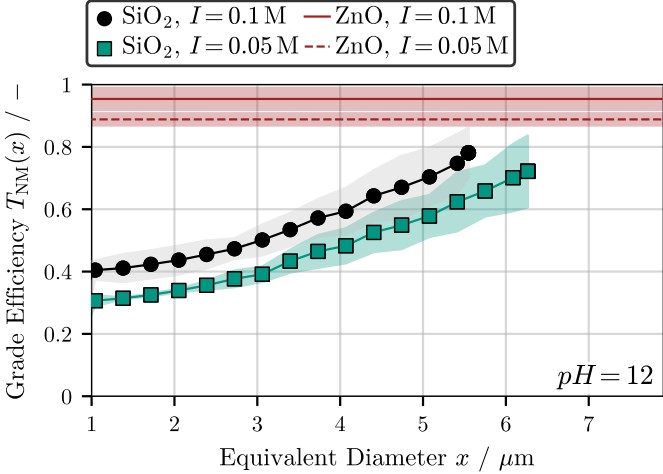

**Figure 9.** Grade efficiency curves for SiO₂ (SF600) and ZnO at constant $pH$ and varied ionic strength. Only the overall separation efficiency $A_{\mathrm{NM}}$ of ZnO was measurable and is shown as a constant horizontal line. The shaded areas represent the standard deviation of triple determination.

### 3.5. Notes on Selective Separation Based on Hydrophobicity

The findings from Section 3.2 show that heteroagglomeration processes between nonmagnetic particles lead to a loss of selectivity in multicomponent systems. When separation is based on surface charge, this effect can be mitigated by choosing appropriate suspension parameters, but never completely prevented, because agglomeration is induced by a *dissimilarity* in the separation criterion. If magnetic (M) and nonmagnetic particles of material 1 (NM1) are oppositely charged, agglomeration takes place. At the same time, the second nonmagnetic component NM2 should be selectively excluded from the agglomerates, which is why M and NM2 must exhibit a similar surface charge. However, this results in the fact that an affinity between NM1 and NM2 cannot be avoided, since NM1 and NM2 also differ in charge. Selective separation based on hydrophobicity is expected to mitigate selectivity loss, since it is induced by a *similarity* of the surface. A strong and long-ranged hydrophobic interaction was shown between two similar, hydrophobic particle surfaces [58,59], while dissimilar surfaces, i.e., one hydrophilic and one hydrophobic surface, were shown to exhibit either no or no long-ranged hydrophobic attraction [59,60]. Therefore, an agglomeration tendency between M and NM1 does not directly result in agglomeration between NM1 and NM2, and selectivity is preserved. It should be noted that the discussion about hydrophobic particle–particle interactions is ongoing, and no overall theory explaining all measured effects was proposed as yet. Currently, the most promising explanation lies in the bridging force of nanobubbles on the particle surface [61,62].

A previously published study [34] put this hypothesis to the test by investigating selective separation between hydrophilic cellulose and hydrophobic microplastic particles. It was shown that cellulose is indeed only separated to a negligible extent even in the multicomponent suspension. The investigated hydrophobic microplastic particles, on the other hand, showed significantly higher separation efficiencies, and some polymers were even separated completely, resulting in high selectivities. The results indicate that separation efficiencies from individual separation experiments can be transferred almost unchanged to the multicomponent system. This suggests that no agglomeration between the nonmagnetic particles took place and underlines the benefits of hydrophobicity as separation criterion for selective separation.

### 3.6. Notes on Breakup and Recycling of Magnetic Seed Particles

The necessary addition of magnetic seed particles is one of the main drawbacks of MSF. In order to realize the process in a sensible way from an economic and ecological point of view, recycling and reuse of the magnetic fraction is unavoidable. Depending on the application, the separated nonmagnetic fraction enclosed in the agglomerates may also be a value product and requires recovery after the process. In a previously published study [46], three different agglomerate processing and seed particle recycling strategies were tested experimentally, which all have individual advantages and areas of application: During thermal breakup, the nonmagnetic fraction is decomposed at elevated temperatures, being applicable in clarification processes where mostly only a particle-free fluid is desired. During chemical breakup, the nonmagnetic particles are dissolved in a solvent, which may, e.g., be suitable for effluent treatment of polymer production plants. Additionally, agglomerates can be broken up mechanically by applying shear forces, which is mainly beneficial if both the magnetic and nonmagnetic fraction should be obtained in their original, particulate form. Through investigation of all approaches in a cyclical manner, eventual long-term effects on the magnetic component were disclosed. In general, all approaches showed high recycling rates of the magnetic fraction. In the case of thermal and mechanical recycling, this results in consistently high separation efficiency over the course of multiple cycles, which underlines that the surface properties of the recovered magnetic particles remain intact. Additionally, in the case of mechanical recycling, a large fraction of the separated nonmagnetic material was recovered. Chemical breakup showed decreasing separation efficiencies, which was retraced to an accumulation of polymer in the recycled magnetic material. However, by varying the experimental parameters, this

effect was shown to be partially reversible. In summary, the feasibility of all three recycling approaches was proven experimentally, and their respective advantages, limitations, and potential applications were discussed.

### 3.7. Notes on Modeling of Heteroagglomeration Processes

The results of Section 3.4 raise the question of what happens in the transition from three-component to arbitrary multicomponent systems. However, Section 2.7 underlines that multidimensional analytical methods are either laborious or not available at all, especially for an increasing number of components. In addition, the analytical tools are limited to the pre- and postseparation state, while the defining heteroagglomeration remains inaccessible. Both issues motivate a parallel theoretical investigation of the MSF process. In previous studies, a discrete population balance model (PBM) has been developed and coupled with a magnetic separation criterion. In principle, this allows the modeling of agglomeration processes in a one-, two-, or three-component system. Two major challenges emerged in this regard: The first is how to calculate agglomeration rates between heterogeneous agglomerates composed of different materials. A simple averaging of the surface-specific properties leads to nonphysical results and is not suitable for this purpose. Therefore, in a previous publication [63], the so-called *collision case model* was developed and validated, which allows the estimation of the desired kinetic parameters only on the basis of the (known) agglomerate composition and the properties of the primary particles. The second challenge is the availability and accuracy of material and process parameters. Required HAMAKER constants can be off by orders of magnitude due to surface roughness [64], and the use of zeta potentials is by definition only a proxy for the unmeasurable surface potential. These uncertain values are then used in assumption-laden model equations to derive the kinetic parameters $\alpha$ and $\beta$. All these uncertainties add up and ultimately result in the fact that purely predictive modeling is impossible and that experimental studies are always required to calibrate the model used. A recently published study [56] shows a promising way to do this by integrating data-driven models. The PBM (white-box) is extended in different ways with machine learning algorithms (black-box), and the resulting hybrid models are able to increase model accuracy while reducing model complexity.

This toolbox should be used in further studies to investigate multidimensional separation; i.e., the model should be extended and applied to arbitrary multicomponent systems. This could help to optimize for selectivity and gain further insight into the process-determining microprocesses.

## 4. Conclusions

This work is concerned with advancing the lesser-known magnetic seeded filtration (MSF) into a selective and multidimensional solid–liquid separation process. During MSF, magnetic seed particles are added to a suspension, and after selective heteroagglomeration with nonmagnetic target particles, magnetic separation of the agglomerates is performed. Both particle–particle interactions and flow effects are shown to drastically influence agglomeration behavior.

The $pH$ value controls the surface charge of the particles and was identified as a main parameter. In the case of a strong, similar charge, repulsive electrostatic interactions counteract agglomeration between particles and consequently separation. However, by increasing the ionic strength, the DEBYE length and consequently the range of this interaction can be reduced to such an extent that the attractive van der Waals forces prevail and agglomeration occurs. A shift of the $pH$ value towards the isoelectric point leads to generally weakened electrostatics, increased agglomeration rates, and ultimately an increase in separation efficiency. In a multicomponent system, i.e., when multiple nonmagnetic systems are present, the focus lies predominantly on the selectivity of the process. This work provides an as yet lacking definition of selectivity for solid–liquid separation processes that is based on relative separation efficiencies between target particles. It was shown experimentally that a selective separation based on surface charge is possible, but the selectivity is below

the expectations based on individual separation experiments. This is due to the fact that separation is based on a dissimilarity in surface charge, which also results in an affinity between the nonmagnetic particles. This subsequently leads to the formation of multicomponent agglomerates, the undesired separation of all components, and the observed loss of selectivity. Multidimensional separation requires the dependence of separation efficiency on a second particle property. This work showed that the grade efficiency is increased with increasing particle size. An in-depth discussion revealed that this seemingly trivial dependence is the result of many opposing effects: Increasing particle size increases the collision frequency between particles, but at the same time reduces the probability that these collisions lead to agglomeration. Since the balance of both effects strongly depends on process conditions, a generalizing statement on how separation is influenced by particle size is not permissible. Finally, all experimental findings were combined and multidimensional separation experiments of a multicomponent suspension of ZnO and $SiO_2$ performed. Again, ZnO was selectively separated, while the grade efficiency of $SiO_2$ was increased for larger particles. The experimental results, therefore, prove that a simultaneous separation based on surface charge and on particle size, i.e., a multidimensional separation, takes place. Consequently, the mixture is classified into a fraction containing all ZnO as well as predominantly large $SiO_2$ particles and a fraction containing only the fine fraction of $SiO_2$ particles.

It remains to be seen whether and how selectivity can be achieved in a complex multi-component suspension, i.e., in the presence of a large number of non-=magnetic particle systems. This work clearly shows that it is essential to avoid unwanted heteroagglomeration between the nonmagnetic particles. In a previous study [34], hydrophobic interactions were found to mitigate this problem and, therefore, indicate a promising way forward. Modification of particle surface by targeted adsorption of surfactants should therefore be investigated in future studies. Particle size selectivity also requires further optimization, especially with respect to multidimensional separation. Since the interplay between collision frequency and efficiency is crucial, specific perturbations, e.g., by varying the shear rate, may lead to sharper separation and should be investigated. In general, the results of this work show that even small amounts of magnetic material are sufficient to achieve high degrees of separation even in dilute suspensions. In addition, a previous study [46] showed that the magnetic particles can be effectively recovered. Questions regarding the scale-up and industrial application of MSF should therefore focus less on the cost of magnetic seed material and more on whether the multilayered agglomeration processes can be realized analogously in a less defined industrial environment. The implementation of a continuous process should also be investigated, with special emphasis on ensuring reliable particle dispersion and residence time in the system. Nevertheless, MSF shows distinct advantages over established separation techniques, as it is suitable for both dilute suspensions and small particle sizes. This work shows that MSF is able to achieve high overall separation efficiencies, while simultaneously being able to selectively separate with respect to either surface charge, particle size, or even both during multidimensional separation.

**Author Contributions:** Conceptualization, F.R.; methodology, F.R.; formal analysis, F.R., O.Z. and E.S.; investigation, F.R., O.Z. and E.S.; data curation, F.R.; writing—original draft preparation, F.R.; writing—review and editing, F.R.; visualization, F.R.; supervision, H.N.; project administration, H.N.; funding acquisition, H.N. All authors have read and agreed to the published version of the manuscript.

**Funding:** This research was funded by the DFG (German Research Foundation) in the priority program 2045 Highly specific and multidimensional fractionation of fine particle systems with technical relevance with grant numbers NI–41432–1 and NI–41431–2.

**Institutional Review Board Statement:** Not applicable.

**Informed Consent Statement:** Not applicable.

**Data Availability Statement:** The presented datasets are available from the corresponding author on reasonable request. Zeta potentials and particle size distributions were published together with experimental data of Section 3.1 in open access [57].

**Conflicts of Interest:** The authors declare no conflict of interest. The funders had no role in the design of the study; in the collection, analyses, or interpretation of data; in the writing of the manuscript; or in the decision to publish the results.

**Disclaimer:** Methods, experimental data, figures, and discussions were previously published in German as part of the following dissertation work (thesis): Rhein, F.. Mehrdimensionale Trennung mit Hilfe magnetischer Partikel. Dissertation, Karlsruhe Institute of Technology (KIT), Karlsruhe Germany, 10 June 2022 [65].

## Abbreviations

The following abbreviations are used in this manuscript:

| | |
|---|---|
| AUC | analytical ultracentrifugation |
| IEP | isoelectric point |
| M | magnetic component |
| MSF | magnetic seeded filtration |
| NM | nonmagnetic component |
| PBM | population balance model |
| PSD | particle size distribution |

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
