# Peer review of "Multidimensional Separation by Magnetic Seeded Filtration: Experimental Studies"

_2674-0516, doi:10.3390/powders2030037_

Round 1

Reviewer 1 Report

Magnetic seed separation has been widely applied in the rapid purification of waste water and other liquids, including selective separation of solid suspended particles, colloids, ions, and other components. The manuscript investigated the influence of factors such as pH value, ion intensity, and particle size on the efficiency in the process of ZnO and SiO2 particles separation using magnetic seed filtration method. Although it provided some reference value, the innovation is limited, specifically in the following aspects: 

1. What is the magnetic induction intensity for separation after the magnetic seeds and non-magnetic mineral agglomeration? The manuscript did not provide an explanation, while the magnetic induction intensity significantly affects the carry-over effect of non-agglomerated particles during the separation process. 

2. Were SiO2 and ZnO particles obtained from natural minerals or artificially synthesized? The manuscript did not present XRD data or provide an explanation, and there can be significant differences in the surface properties of particles from these two sources. 

3. SiO2, ZnO, and SiO2-MAG particles all exhibited negative surface charges at pH > 8. How did selective agglomeration occur without altering the surface charge properties? The phenomena and mechanisms provided by the manuscript were insufficient. It would be helpful to include SEM images after agglomeration. 

4. The manuscript mentioned that particle size affected the agglomeration separation efficiency. According to other researchers, the particle size of the magnetic seed had a significant influence on magnetic agglomeration and separation due to its dominant role in the magnetic separation process. It is recommended that the author supplement the corresponding research in this regard. 

5. Currently, the main challenge in magnetic seed separation remains the selective agglomeration of magnetic seeds. In practical separation systems, the differences in electrical properties or surface charges among the components are not significant enough to achieve separation solely by adjusting the pH value. Therefore, it is necessary to add surfactants to selectively adsorb onto the surface of a specific component, making it easier to agglomerate with magnetic seeds. Alternatively, surface modification of the magnetic seeds can be done by adding targeted molecular groups that enable selective adsorption and agglomeration with a specific component. Further research is needed in the manuscript.

Author Response

What is the magnetic induction intensity for separation after the magnetic seeds and non-magnetic mineral agglomeration? The manuscript did not provide an explanation, while the magnetic induction intensity significantly affects the carry-over effect of non-agglomerated particles during the separation process.

This is a valid point that is insufficiently discussed in the original manuscript. In section 2.3. lines 174ff. the description of the magnetic separation process is extended with the following discussion. The magnetic flux density of the used permanent magnet is given (with respective reference). It should be noted that the local magnetic field gradients inside the suspension, i.e. next to the ferromagnetic mesh, were not measurable. Therefore, preliminary studies were performed that guaranteed full separation of pure magnetic suspensions, while purely non-magnetic suspensions did not show any measurable separation efficiency. This shows that 1) magnetic field gradients are sufficiently high and 2) that separation of non-magnetic particles can be entirely attributed to the hetero-agglomeration.

Were SiO2 and ZnO particles obtained from natural minerals or artificially synthesized? The manuscript did not present XRD data or provide an explanation, and there can be significant differences in the surface properties of particles from these two sources.

More information on the non-magnetic particles is provided in section 2.2 lines 140ff. Both systems are purchased from Quarzwerke GmbH and are produced from prepared natural raw minerals (as stated in the product data sheet). Although interesting, XRD data was not measured as the authors feel that presented zeta potential and particle size distribution data is sufficient for the context of this work.

SiO2, ZnO, and SiO2-MAG particles all exhibited negative surface charges at pH > 8. How did selective agglomeration occur without altering the surface charge properties? The phenomena and mechanisms provided by the manuscript were insufficient. It would be helpful to include SEM images after agglomeration.

Generally, this effect is discussed in chapter 2.1. and 3.1. 273ff.: The absolute value of the zeta potential is decisive of whether agglomeration occurs or not, as there is an interplay between attractive van der Waals and repulsive electrostatic forces. Based on the results from section 3.1, it is understandable, that ZnO is selectively separated as these particles have a lower zeta potential than SiO2 and therefore lower electrostatic repulsion towards the magnetic particles. This discussion was added to section 3.2. in lines 300ff. to improve the clarity of the discussion.

The manuscript mentioned that particle size affected the agglomeration separation efficiency. According to other researchers, the particle size of the magnetic seed had a significant influence on magnetic agglomeration and separation due to its dominant role in the magnetic separation process. It is recommended that the author supplement the corresponding research in this regard.

It is true that particle size influences the magnetic force and therefore magnetic separation efficiency. The discussion on this was extended in chapter 3.3. lines 376ff. with the following discussion:

Concerning the magnetic separation, Eq. 8 shows that an increasing magnetic particle size results in a higher absolute magnetic force per particle. However, as inertia during separation is also increased, an earlier study [56] derived a separation criterion that showed that magnetic separation of a hetero-agglomerate is mainly influenced by the volume ratio between the magnetic and non-magnetic component. It should be noted that in this study the size of the magnetic seed particles was not varied, however it is sensible to assume that larger non-magnetic particles are more likely to form agglomerate with a lower magnetic partial volume and might therefore be separated to a lesser extent. Nevertheless, the influence of particle size on the separation efficiency is ultimately determined by the agglomeration kinetics:

On the one hand, …

Thank you for emphasizing this point. The authors hope that the addition of this discussion sheds some light into the intertwined dependencies of agglomeration kinetics and magnetic separation performance. As only the size of the non-magnetic component was varied in this study, the main influence lies – in our opinion – in the agglomeration kinetics rather than the magnetic separation.

Currently, the main challenge in magnetic seed separation remains the selective agglomeration of magnetic seeds. In practical separation systems, the differences in electrical properties or surface charges among the components are not significant enough to achieve separation solely by adjusting the pH value. Therefore, it is necessary to add surfactants to selectively adsorb onto the surface of a specific component, making it easier to agglomerate with magnetic seeds. Alternatively, surface modification of the magnetic seeds can be done by adding targeted molecular groups that enable selective adsorption and agglomeration with a specific component. Further research is needed in the manuscript.

This is a valid and interesting point that should definitely be investigated in future studies. However, as targeted modification / pre-treatment of particle surface requires detailed and sophisticated investigations, the authors don’t feel the requirement to perform this in the submitted study.

Nevertheless, this idea is integrated into the outlook in lines 575f.  

Reviewer 2 Report

The paper describes a study of magnetic seeded filtration as a multidimensional separation process for particle systems. The study shows that the technique can achieve high separation efficiencies by inducing selective hetero-agglomeration with non-magnetic target particles and subsequent magnetic separation. The results also highlight the need for multidimensional evaluation in particle separation and the potential of magnetic-seeded filtration as a promising technique.

The introduction and the list of references comprehensively reveal the problems and relevance of the study. The problem statement is presented at an accessible level for readers understanding. The authors have clearly defined the direction of the study (size dependence of the separation and the case of a multicomponent system for MSF). The research methods cover the Magnetic Seed Filtration Theory and experimental investigation, including UV-VIS, AUC and multidimensional analysis. The results of the study look logical and do not cause doubt. After the results, the authors provide notes on selective separation based on hydrophobicity, breakup and recycling of magnetic seed particles and modelling of hetero-agglomeration processes. The conclusions summarise the research undertaken and contain all the relevant information presented in the article.

To summarize, the paper is a complete study. There are no significant omissions or shortcomings in the article. The paper is well structured, easy to read and contains all necessary figures at a sufficient level. The article is recommended for publication.

Author Response

No objections

Thank you very much for reviewing the manuscript and the positive feedback.

Reviewer 3 Report

In the manuscript "Multidimensional Separation by Magnetic Seeded Filtration: Experimental Studies" the authors experimentally investigate the applicability of magnetic seeded filtration as a multidimensional separation process. The document is well-written and presents a good structure. Despite my positive comments, some aspects should be reviewed before publication:

Introduction: Figure 1 would be move to Material and Methods. The authors should emphasize only the elements of interest for the study. In addition, the Introduction should specify the novelty of the study. What potential uses does magnetic separation have? Recently, a study was published on magnetic separation applied to ferruginous and titaniferous sands (doi: 10.3390/resources11120121).

Materials and Methods: 2.1. Magnetic Seeded Filtration Theory, is it necessary?. Please move to Introduction. The Materials and Methods is too long. I suggest the authors place a diagram that allows to summarize the study's methodology.

Results and Discussion: It would be interesting to include potential applications of this technique large-scale processes. Comment on aspects of cost with the methodology proposed in this manuscript.

Please improve the Abstract and Conclusions considering all observations.

Minor observation: placing a list of abbreviations at the end of the document is suggested.

n/a

Author Response

Introduction: Figure 1 would be move to Material and Methods. The authors should emphasize only the elements of interest for the study. In addition, the Introduction should specify the novelty of the study. What potential uses does magnetic separation have? Recently, a study was published on magnetic separation applied to ferruginous and titaniferous sands (doi: 10.3390/resources11120121).

This manuscript concludes a research project funded over the course of 6 years and serves as final report. Therefore, the authors (and funders) feel that briefly discussing other aspects than the core results of the presented study is reasonable and gives a more holistic view on the process.

Speaking from experience, describing the MSF process without a simple scheme (as Fig. 1), is rather challenging. Although this figure obviously describes the entire process and would therefore be suitable for the Materials and Methods section, the authors feel that it is required in the Introduction to understand the process at hand.

Magnetic separation is obviously applied in a variety of applications. The recommended work as well as a recent review article are added in lines 72f.

Materials and Methods: 2.1. Magnetic Seeded Filtration Theory, is it necessary?. Please move to Introduction. The Materials and Methods is too long. I suggest the authors place a diagram that allows to summarize the study's methodology.

As this manuscript is concluding a research project funded over the course of 6 years, the authors feel that an extended theory section (used for discussion of the measured effects) and a thorough materials and methods section are reasonable.

Depending on how strict MDPI adheres to its given structure, the theory might be moved to an individual section “2. Magnetic Seeded Filtration Theory”, which should probably be done during production of the final manuscript.

As you mentioned, the Materials and Methods section is already rather long. We do have a graphic representation of the methodology; however, we are unsure whether to add this to the manuscript or not. Especially since the methodology is referenced to https://doi.org/10.1002/cite.201900104, which also contains such a graphic. If the editor feels that this is necessary we will gladly provide a figure.

Results and Discussion: It would be interesting to include potential applications of this technique large-scale processes. Comment on aspects of cost with the methodology proposed in this manuscript.

Obviously, the presented results are performed with model systems and aim at generating a fundamental understanding of the MSF process considering its dependences on size and surface charge. Therefore, discussing specific applications and cost aspects of the scaled-up process are hard to do and should in our opinion not be part of the Results section (especially since they were not investigated).

However, the current state of the conclusion already touches on these aspects in lines 579ff. The fact that high separation efficiencies are achieved even in low concentrations with low material use of magnetic particles (that are also recyclable) paired with the application of permanent magnets makes a scaled-up version of the MSF process cost effective. However, as mentioned in the conclusions, this will be the topic of future research.

Please improve the Abstract and Conclusions considering all observations.

Regarding the conclusion: It is not entirely clear what the reviewer means. The observations regarding multidimensional separation were slightly expanded in lines 564ff, however the authors feel that all discussed results of sections 3.1 – 3.4 are included already. Assuming that the reviewer means the results of chapters 3.5 – 3.7, they are not included in the conclusions because – as mentioned above – they are added to give a more holistic picture of the MSF process and do not constitute “new” findings in the context of this work. Obviously, references are made in the respective subsection and results are more thoroughly discussed in the individual publications.  

The same can be said about the abstract. Considering the MDPI word limit, the authors feels that all main findings of this specific work, i.e. charge based, charge selective, size dependent and multidimensional separation are given, while also providing some motivation and context of this work. Results of chapters 3.5. – 3.7 are not included, again, due to them being added to give a more holistic view that also serves as overall summary of the 6-year project.

Minor observation: placing a list of abbreviations at the end of the document is suggested.

An abbreviation list was added at the end of the manuscript.

Round 2

Reviewer 1 Report

  • Revised manuscripts are acceptable.

Reviewer 3 Report

Thanks for revising the manuscript based on the suggestions provided in the first round.

n/a